# Tunable Fano Resonances in an Ultra-Small Gap

**Fuqiang Yao, Fang Li \*, Zhicong He, Yahui Liu, Litu Xu and Xiaobo Han \***

Hubei Key Laboratory of Optical Information and Pattern Recognition, School of Optical Information and Energy Engineering, School of Mechanical and Electrical Engineering, Wuhan Institute of Technology, Wuhan 430073, China; yfq15872268171@163.com (F.Y.); hzc_900503987@163.com (Z.H.); 13212799763@163.com (Y.L.); 15871430579@163.com (L.X.)

**\*** Correspondence: lifang@wit.edu.cn (F.L.); hanxiaobo@wit.edu.cn (X.H.)

**Abstract:** A Fano resonance is experimentally observed in a single silver nanocube separated from a supporting silver film by a thin aluminum oxide film. The resonance spectrum is modulated by changing the size of the silver nanocube and its distance from the silver film. The system is fabricated by a bottom-up process with an accurately controlled nanogap at the sub-6-nm scale. The simulation result shows that the destructive interference between the dipole mode and the quadrupole mode in this "nanocube on mirror" (NCoM) structure is responsible for the resonance. The spectra red-shifted as the size of the silver nanocube increased and its distance from the silver film decreased. In addition, a refractive index sensitivity of the spectrum of 140 meV/RIU (refractive index unit), with a 2.4 figure of merit, is obtained by changing the dielectric environment around the silver nanocube. This work will enable the development of high-performance tunable optical nanodevices based on NCoM structures.

**Keywords:** Fano resonance; interference; nanocube; sensitivity; surface plasmon resonance

## 1. Introduction

When discrete and continuous excited-state energy levels overlap, quantum interference occurs. A zero absorption occurs in a specific optical band, and thereby generates an asymmetric spectrum that is called a Fano resonance [1]. It was first reported by U. Fano when studying atomic spectra, and he theoretically attributed the asymmetric resonance to a destructive interference that could not be described by the Lorentz formula. The destructive interference suppresses the radiation attenuation of the system, which also greatly increases its sensitivity to the surrounding media and concentrates the system energy on the surface of the structure [2]. Relative to the initial surface plasmon mode [3–5], the Fano resonance has a stronger near-field enhancement and a finer spectrum, as well as a higher figure of merit (FOM). Therefore, it has potential applications in biosensing and field enhancement [6]. Other potential applications include lasers [7], optical switches [8,9], nonlinear optics [10,11], high-quality-factor devices [12], slow-light devices [13], and surface-enhanced Raman scattering [14].

Fano resonances in metallic nanostructures have also been observed in optical responses [15,16]. These nanostructures have abundant localized surface plasmon resonances (LSPR) that originate from collective electron oscillations on the metal surface. Common LSPR modes include dipoles, quadrupoles, and octupoles. The dipole mode has a large dipole moment, with a large resonance bandwidth in the spectrum, while the quadrupole and the octupole modes have small dipole moments, with small resonance bandwidths in the spectrum. Radiation from the broad resonance mode has high attenuation but can be coupled as a "bright mode" into an incident electromagnetic field. Meanwhile, a narrow resonance mode with a small net dipole moment has very low radiation attenuation and cannot be coupled into an incident electromagnetic field; this is the "dark mode" [17,18]. The presence of bright and dark modes gives rise to Fano resonances in metallic nanostructures. In complex

structures, there are abundant plasmon hybridizations among various plasmon modes that lead to Fano resonances, such as a disk-ring nanocavity [19], a metal core-shell nanostructure [20–22], a self-assembled metal nanostructure [6,14,23,24], a metal nanoparticle (NP) array [25–28], and a waveguide and cavity resonator coupling structure [29,30]. These complex structures usually require complicated fabrication methods. For example, self-assembled metallic nanostructured pentamers require electron beam lithography, which is not only costly, but also involves highly controlled precision and considerable time [23]. It would be advantageous to fabricate simple metal nanostructures at low cost to realize Fano resonances.

Recently, Fano resonances were observed in silver nanocube dimers [31,32], where small spacer separations occur via surfactants. However, precise control of these small nanogaps is challenging and the silver nanocube dimers are easily misaligned. Thus, they are poorly tuned and difficult to integrate. A metallic substrate supporting metallic NPs forms a NP-on-mirror (NPoM) structure, which breaks the symmetry [33,34]. The presence of the substrate creates a non-uniform dielectric environment around the metallic NPs, enabling plasmon mode hybridization that couples broad and narrow plasmon resonances. When the size of the metal NP is large, dipole mode and multipole mode are generated, but these modes will not be coupled with each other. When the dielectric environment around the metal NPs changes, such as placing a substrate near the metal NPs, it creates conditions for interference between the modes. The destructive interference between the dipole mode and the higher-order mode will generate Fano resonance. Due to the presence of the metal film in the NPoM structure, a mirror image of metal NPs is formed in the metal film, which is similar to the dimer structure. Moreover, the distance between the NPs and the substrate can be easily controlled by varying the thickness of an intervening dielectric layer, especially when the gap is reduced below 10 nm, a strong local field will be formed in the gap of the NPoM structure. Since the NPoM structure consists of a metallic film, a dielectric layer, and single NPs, it usually does not require costly fabrication techniques [35–41]. Therefore, it is expected to observe the Fano resonance in NPoM structures.

Here, a Fano resonance is observed in a silver-nanocube-on a-silver-mirror (NCoM) structure that was fabricated by a bottom-up process with precisely controlled sub-6-nm gaps. The Fano resonance position is modulated by changing the nanocube size and its gap distance from the silver film. To examine the origin of the Fano resonance, the electric field magnitude profiles and charge distribution of the NCoM structure are numerically simulated; a mode analysis then reveals that the scattering spectrum is a Fano resonance. The refractive index sensitivity of the Fano resonance spectrum in the NCoM structure is also investigated by varying the dielectric environment around the silver nanocube.

## 2. Experimental Structure and Model Parameters

A schematic of the NCoM structure is depicted in Figure 1a. It consists of a single silver nanocube and a silver film. An aluminum oxide ($Al_2O_3$) spacer layer lies between the nanocube and the silver film. Figure 1b is a cross section of a NCoM structure prepared on a silicon substrate, and Figure 1c is a transmission electron microscope (TEM) image of silver nanocubes that reveals their cubic shape. The radius of the rounded corner is defined as $r$, and the face-to-face width of each nanocube is defined as $d$. The NCoM structure was fabricated in a bottom-up process. Firstly, an 80-nm-thick silver film was coated on a silicon substrate via electron beam evaporation. Then, the $Al_2O_3$ spacer film was deposited on the silver film via atomic layer deposition at 100 °C. Finally, a silver nanocube solution was dropped onto the $Al_2O_3$ film, which was rinsed with deionized water after 1 min to remove excess nanocubes. The NCoM sample was then blown dry with nitrogen gas. The silver nanocubes were uniformly monodispersed on the $Al_2O_3$/silver film.

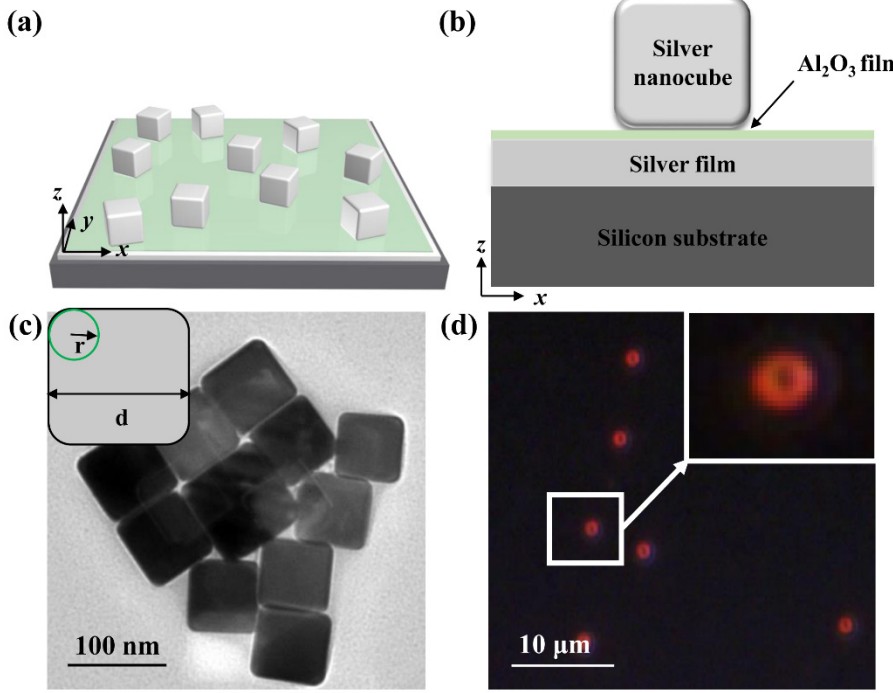

**Figure 1.** (**a**) 3D diagram of the NCoM structure, where silver nanocubes are randomly distributed on the Al2O3 film; (**b**) Cross-section diagram of the NCoM structure prepared on a silicon substrate, consisting of silver nanocubes on a silver substrate, separated by the Al2O3 film; (**c**) Transmission electron microscope (TEM) image of the silver nanocube. The insert is a cross section of a single silver nanocube: r is the radius of the rounded corner size, and d is the face-to-face width of the silver nanocube; (**d**) Dark-field optical image of the silver nanocubes.

The experiments were performed at the single-particle level with a dark-field scattering microscope (Olympus, BX 53). The scattering spectrum is obtained by irradiation with a halogen lamp without any polarizer added. A dark-field objective (Olympus, 100×, NA = 0.9, MPLFLN) was used to illuminate the samples and collect the scattered light, which was sent to a CCD(charge coupled device) (Qimaging, QICAM B series) or a spectrometer (Andor, SR303i). All the experiments were performed at room temperature. Figure 1d is a dark-field image of the NCoM structure, which is likely a Laguerre Gaussian 01 donut [37]. Since the average distance between the individual nanocubes was greater than 3 μm, interactions among them were negligible.

Numerical calculations were performed with a commercial finite difference time domain (FDTD) Solutions 8.19.1584, which was developed by Lumerical Solutions. All the sharp corners (edges) of the nanocubes are slightly smoothed by the spherical (cylindrical) surface, consistent with experimental observations. The silver nanocube and its surrounding space are divided into 0.5 nm meshes. The incident light illuminates the metal surface vertically, and the polarization of the electric field is along the x-axis. The refractive index of silver is referred from the commonly used data from Johnson and Christy [42]. As the surface of each nanocube is covered by surfactant molecules (Polyvinylpyrrolidone, PVP), the nanocubes are located 1 nm above the $Al_2O_3$ film for the simulation. The specific structural model parameters are listed in Table 1.

**Table 1.** Structural model parameters of numerical simulation.

| Structure | Parameters |
|---|---|
| The face-to-face width of the cube ($d$) | 75 nm |
| The rounded corner size ($r$) | 10 nm |
| The thickness of the $Al_2O_3$ film | 2 nm |
| The thickness of the silver film | 80 nm |

## 3. Results and Discussion

### 3.1. Observation and Proof of Fano Resonance

When the nano-gap thickness is greater than 10 nm in the NPoM structure, the scattering spectrum shows one resonance peak. While the thickness decreases and is less than 10 nm, this resonance peak will be a gradual red-shift and splits into two resonance peaks [41]. As shown in Figure 2, when $d$ = 72 nm, $r$ = 9 nm, and the nano-gap thickness is 4 nm, two scattering resonance peaks are clearly observed with wavelengths of 588 nm (peak II) and 640 nm (peak I). From the profile view of the spectrum, the scattering spectrum shows an asymmetric shape. To investigate the origin of this asymmetric shape, the electric field magnitude profiles of peak II and peak I on the x-z plane are given in Figure 2b,c, respectively. Figure 2d,e are charge distribution diagrams corresponding to Figure 2b,c, respectively. The silver nanocube-$Al_2O_3$ film-silver film structure forms a plasmonic nanocavity. In Figure 2b,c, there is a "hot spot" on the left and right sides of the nanocavity. It indicates that the electric field profile follows the pattern of the charge distribution on the bottom of the nanocube with two areas of intense field at either side of the nanocube. Compared to Figure 2b, the top surface of the cube in Figure 2c has stronger field enhancement, although both are smaller compared to the bottom surface of the cube. From the charge distributions in Figure 2d,e, peak II is a quadrupole mode and peak I is a dipole mode. This charge distribution pattern is related to the substrate. The substrate effect generally has two mechanisms: the pure screening effect and the substrate mediated interaction of the initial cube plasmons. The former is the image charge generated by the plasmon mode acting on itself, and the latter is the interaction with other plasmon modes [33]. The dipole mode and the quadrupole mode interfere with each other under the action of the substrate, thereby generating the Fano resonance in the scattering spectrum of the NPoM nanostructure.

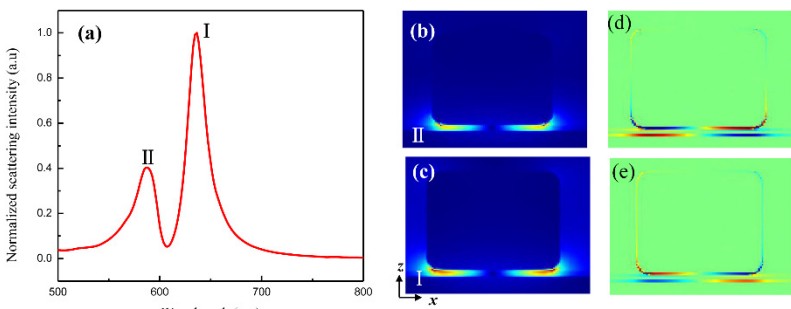

**Figure 2.** (**a**) Scattering spectrum of a single silver nanocube on a 4-nm spacer layer. (**b,c**) The electric field magnitude profiles of peak II and peak I on the x-z plane, respectively. (**d,e**) The plasmon charge distribution of peak II and peak I on the x-z plane, respectively.

### 3.2. Effect of Nanocube-Substrate Spacing on Fano Resonance

In the experiment, four different $Al_2O_3$ film thicknesses were prepared (2 nm, 3 nm, 4 nm, and 6 nm). Figure 3a shows the experimental scattering spectra of NCoM structures with the different gaps. When the gap thickness was 6 nm, the NCoM scattering spectrum had only one dominant peak at 640 nm over the range 500–800 nm. The broadening of the scattering spectra is mainly caused by the loss of the metal itself. An unsmooth metal surface can increase the loss, and lead to spectral broadening.

When the gap was reduced to 4 nm, the scattering spectrum had two resonance peaks at 622 nm and 672 nm, with a dip at 662 nm. When the gap is reduced to 3 nm and 2 nm, the scattering spectra also have two resonance peaks, and the Fano depletion dips and red-shifts to 672 nm and 691 nm, respectively. This red-shift is caused by the damping of plasmon hybrids that is associated with the coupling strength between the NP and the film, which has been well studied [43–45]. In the NPoM structure, the coupling strength between the NP and the film is enhanced as the gap decreases. When the gap is less than 4 nm, the observed dip spectra indicate that hybridization between different resonance modes mediated by the substrate is a near-field interaction. In Figure 3b, numerical simulations indicated a good agreement with the experimental results in Figure 3a. However, the experimental linewidths are larger than the calculated linewidths, which may be caused by the high loss from unsmooth metal film

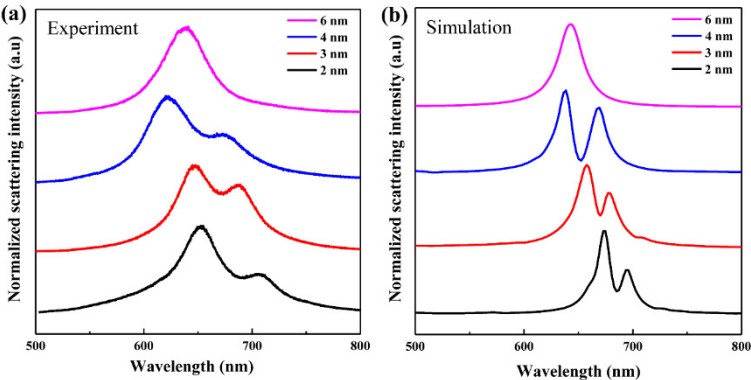

**Figure 3.** Experimental (**a**) and numerically simulated (**b**) scattering spectrum of individual silver nanocubes with different spacer thicknesses: 2 nm, 3 nm, 4 nm, 6 nm, respectively.

### 3.3. Effect of Nanocube Size on Fano Resonance

Figure 4a shows three scattering spectra from difference sized silver nanocubes acquired with the same 2 nm gap thickness. In Figure 4b, a series of simulated scattering spectra resembling the experimental data are obtained by changing the size of the silver nanocube. The simulated radii *r* of the rounded nanocube corners are 9 nm (blue), 9 nm (red), and 10 nm (black), and the corresponding side lengths *d* are 73 nm, 74 nm, and 75 nm, respectively. As the nanocube size increases in the NCoM structure, the Fano depletion dips and red-shifts. When the particle sizes become larger, the distance between the charges on the two opposite surfaces becomes larger. The Coulomb effect between them is weakened, which results in a decrease in the LSPR frequency and the red-shifted of spectra [46]. Therefore, when the nanocube size increases, the dip position of the spectra is red-shifted. Therefore, the Fano resonance in the NCoM structure is very sensitive to the nanocube size.

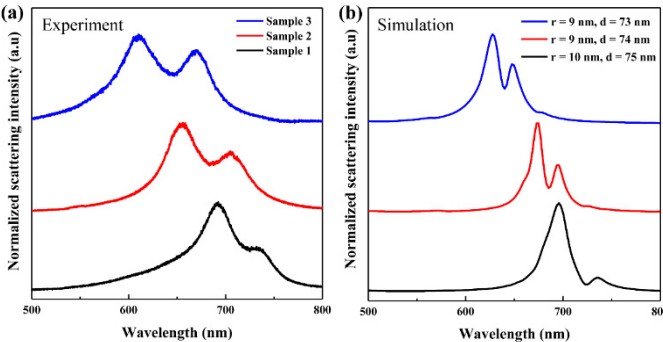

**Figure 4.** (**a**) Scattering spectrum of three different samples in experiment; (**b**) Theoretically simulated scattering spectrum of silver nanocubes with three different sizes.

In addition, we also studied the similar NPoM structure of silver nanocubes with a side length of 100 nm and gold nanospheres with a diameter of 100 nm under the same experimental condition. However, no mode splitting was obtained. It shows that only the specific shape and the appropriate ratio of side length to gap could generate Fano resonance. Moreover, the rounded corner size of the cube has an impact on its scattering spectrum, which is related to the effective mode volume of the plasmonic nanocavity.

*3.4. Effect of Nanocube Dielectric Environment on Fano Resonance*

The dark mode in the Fano resonance refines the spectrum and increases the refractive index sensitivity of the LSPR. Some recent studies have been working to improve refractive index sensitivity [29,30,47]. Here, silver nanocubes were tested in different dielectric environments, including air (*n* = 1.0), water (*n* = 1.33), alcohol (*n* = 1.362), and dimethylformamide (DMF, *n* = 1.428). In the scattering spectra plotted in Figure 5a, the dip position is red-shifted when the refractive index of the surrounding dielectric environment increased. When the size of metal NPs is much smaller than the wavelength of incident light, the calculation model of metal NPs can be replaced by the model of spherical particles, and the LSPR frequency is defined as

$$\omega_{LSPR} = \frac{\omega_P}{\sqrt{2\varepsilon_d + 1}} \qquad (1)$$

where $\omega_p = \sqrt{\frac{N_e{}^2}{\varepsilon_o m_e}}$ is the resonance frequency of plasmon, $N$ is the free electron concentration, $e$ is the electronic charge, $\varepsilon_0$ is the dielectric constant of free space, $m_e$ is the effective electronic mass, and $\varepsilon_d$ is the dielectric constant of the surrounding environment. Therefore, as the refractive index increased, the LSP resonant frequency decreased, resulting in a spectral red-shift. A linear fit (red line) of the Fano resonance shift (black circles) as a function of refractive index $n$ is shown in Figure 5b. The resulting sensitivity is about 140 meV/RIU(refractive index unit), with a FOM value of the Fano resonance of 2.4. These data can provide a reference value for a sensor based on Fano resonance refractive index sensitivity.

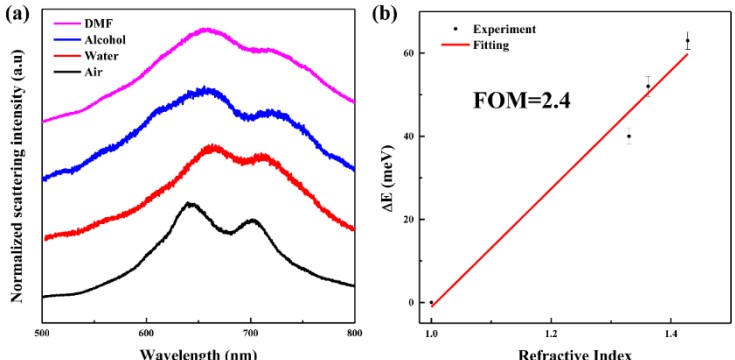

**Figure 5.** (**a**) Scattering spectrum of the same silver nanocube in four different dielectric environments: air (*n* = 1), water (*n* = 1.33), alcohol (*n* = 1.362), and dimethylformamide (DMF) (*n* = 1.428); (**b**) Linear fit of Fano resonance shift as a function of refractive index n. The black spots indicate the experimentally Fano dip position. The red line is a linear fit, which estimated the sensitivity as 142 meV/RIU.

## 4. Conclusions

When the nanogap is reduced to less than 6 nm, the plasmon mode hybridization between broad and narrow resonances results in an asymmetric Fano line shape in the spectra. The NCoM structure is simple to fabricate, and its spectra could be modulated by changing the morphology, the silver NP size, and the Al$_2$O$_3$ spacer nanogap. In both experiments and numerical simulations, the wavelength of the Fano depletion dip strongly depends on the gap thickness and the nanocube size. Additionally,

when the refractive index of the surrounding dielectric environment increases, the dip position is also red-shifted. The resulting sensitivity is 140 meV/RIU, and the calculated FOM value is 2.4. Overall, these results indicate that NCoM nanostructures are a simple and effective platform for examining Fano resonances. It is anticipated that these results could be used for plasma sensors based on Fano resonances in metal nanostructures. Moreover, since the dip of Fano resonance in the NCoM structure is sensitive to the gap thickness, the system can be used to determine the gap thickness. The dielectric layer in the NCoM structure can be replaced with a temperature-sensitive dielectric layer. Thus, the reversible transformation between the Fano line type and the Lorentz line type of scattering spectra can be achieved in the scattering spectrum, which is similar to a temperature-controlled switch.

**Author Contributions:** Formal analysis, F.Y.; Funding acquisition, L.X.; Investigation, F.Y., Z.H. and Y.L.; Methodology, X.H.; Software, Z.H. and L.X.; Supervision, F.L. and X.H.; Writing—original draft, F.Y. and Y.L.; Writing—review & editing, F.L. and X.H. All authors have read and agreed to the published version of the manuscript.

**Funding:** This research was funded by Programs 973 (2014CB92130).

**Acknowledgments:** We thank Jun Zhou for $Al_2O_3$ film fabrication. Special thanks to the Analytical and Testing Center of HUST, the Center of Micro-Fabrication and Characterization (CMFC) and the Center for Nanoscale Characterization & Devices (CNCD) of WNLO for using their facilities.

**Conflicts of Interest:** The authors declare no conflicts of interest.

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
