# Peer review of "Tunable Fano Resonances in an Ultra-Small Gap"

_applsci, doi:10.3390/app10072603_

Round 1

Reviewer 1 Report

The authors provide a very clear experimental and numerical simulation of the optical absorption behaviour of a cube of silver
sitting a precisely spaced distance above a silver film.

There are a handful of points that it would be good for the authors to clarify.

1) In their numerical modelling the authors don't appear to specify the grid size of the finite difference
time domain (FDTD) modelling. They appear to be making small changes (~1nm in Figure 4b) to the relative position of cube and film and to the size of the cube.
Is the system indeed gridded at this tiny scale? What is the grid size?
Or, is spatial averaging used to account for these small changes more accurately with a coarse grid?

2) The paper does not mention whether the metal's dispersion is taken into account by the commercial FDTD.
They specify that the silver's dielectric constant is calculated, but at a single central wavelength or across the spectrum?

3) They don't specify which commercial tool they use. This is critical for reproducibility.

4) Several features of Figure 2 suggest that it is the electric field magnitude that is shown, not the electric field amplitude.
Dark blue represents zero field inside the cube and film. Yet in 2b and c there is a zero field "amplitude"
between the left and right hand sides. This denotes a sign change as the field amplitude crosses zero.

Therefore if this is a cube with C4 symmetry and we stick with the axes denoted in Figure 1b), a mode can be excited equally along y as it can along x.
These are independent field and charge oscillations.
Think Hermite Gaussian modes 10 and 01.
Figure 1b of Reference 37 seems to support this.
Equally excited by unpolarised light and time averaged, they would most likely resemble a Laguerre Gaussian 01 donut.
Figure 2b)s axes labels x and y are confusing, these should be x and z.

Could imperfections couple and slighty split these modes, leading to the broadening seen between 6nm curves in 3a and b?
Unsmooth metal surfaces is very likely to play a role too.

5) In a system which such obvious C4 symmetry, they don't mention the polarisation settings of the incident excition light.
This isn't mentioned at all in the experimental section.
The polarisation of light at the focus of this 0.9 NA objective contains longitudinal (z-directed) components.
In the inset dark field image of the cube there is a distinctly square shape to the "donut".
This could of course be the colourmap or the image compression.
The light source isn't mentioned is it a halogen lamp or arc lamp (what is its spectral content)?
Do these wide field images contain both spectral features / modes?
The donut isn't perfectly uniform, could this then be the CCD registering two overlapping patterns with different emission intensities.
And excitation strengths.

Again Figure 1b of Reference 37 seems to support this.

6) The accurately controlled gaps in the abstract are sub-6nm, whereas on line 67 these precisely controlled gaps are now sub-10nm.

Typos:

Abstract
line 16:
Numerical simulation results show that...

line 20:
...with a figure of merit value of 2.4.

3. Results:
line 104:
...will be gradually redshifted...
line 111:
...amount of charge of opposite...
line 115
,, double full comma

Reviewer 2 Report

In their manuscript, the authors report experimental investigations of Fano resonances in plasmonic cavity modes realized by silver nano cubes placed close to a silver film with a controlled, sub-6nm gap. The authors describe a bottom up process in which their "nanocube on mirrir" system is assembled - previously, the precise control of these gaps has been challenging. Possible applications are promised in the development of high-performance tunable optical nanodevices.

In general, the manusript is written and structured well. For the non expert, I am missing a little more introdcution to the underlying principles and simple models/scaling of plasmonic cavities in the treated system. Otherwise, the significance of the decribed observations is hard to judge / distinguish from a purely phenomenological point of view. E.g.: How does the ratio of cube side length to gap influence the observations? What are other typical length scales in the cavity influencing the results?

Section 3.1 is fairly hard to read and understand and seems to require significant revision. Similarly, I feel like Fig 2 b-e could make a great tool to explain the basic processes happening in the observed system.

Finally, I would apreciate if the authors could spend a few more sentences on the possible applications of this system to make the science a bit more accessible to non experts.

Upon revision I would recommend publication of the manuscript.
